# Self-playing Adversarial Language Game Enhances LLM Reasoning

**Pengyu Cheng**[1]    **Tianhao Hu**[1]    **Han Xu**[1]    **Zhisong Zhang**[1]
**Yong Dai**[1]    **Lei Han**[3]    **Nan Du**[1]    **Xiaolong Li**[2]

Tencent AI Lab [1]Shenzhen & [2]Seattle   [3]Tencent Robotics X Lab

pengyucheng@tencent.com

## Abstract

We explore the potential of self-play training for large language models (LLMs) in a two-player adversarial language game called Adversarial Taboo. In this game, an attacker and a defender communicate around a target word only visible to the attacker. The attacker aims to induce the defender to speak the target word unconsciously, while the defender tries to infer the target word from the attacker's utterances. To win the game, both players must have sufficient knowledge about the target word and high-level reasoning ability to infer and express in this information-reserved conversation. Hence, we are curious about whether LLMs' reasoning ability can be further enhanced by **S**elf-**P**laying this **A**dversarial language **G**ame (SPAG). With this goal, we select several open-source LLMs and let each act as the attacker and play with a copy of itself as the defender on an extensive range of target words. Through reinforcement learning on the game outcomes, we observe that the LLMs' performances uniformly improve on a broad range of reasoning benchmarks. Furthermore, iteratively adopting this self-play process can continuously promote LLMs' reasoning abilities. The code is available at https://github.com/Linear95/SPAG.

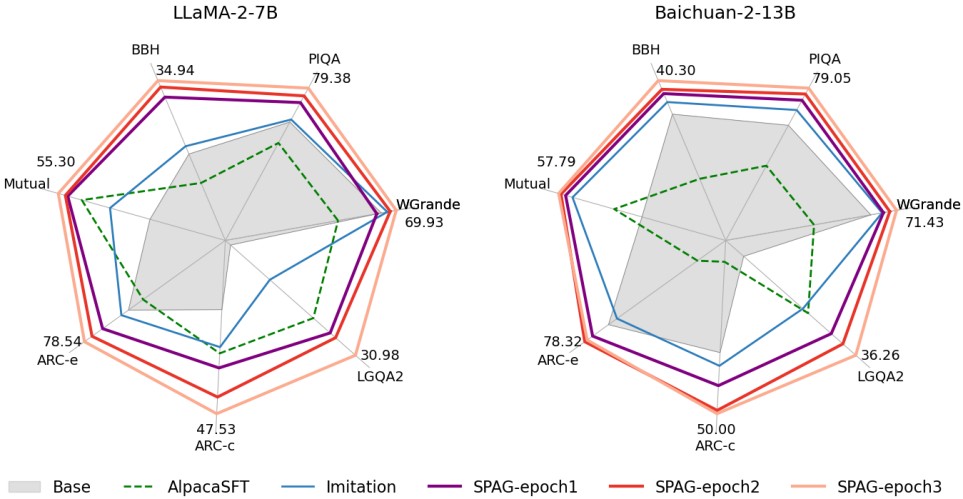

Figure 1: Reasoning improvements from **S**elf-**P**laying of **A**dversarial language **G**ames (SPAG) on comprehensive reasoning benchmarks. With the SPAG epoch increasing, the LLM reasoning ability continuously improves. Each axis is normalized by the maximum answer-accuracy value.

## 1   Introduction

Large language models (LLMs), such as GPT-4 [OpenAI, 2023b] and Gemini [Team et al., 2023], have reformed the domain of artificial intelligence (AI) with astonishing language capacities, such as

38th Conference on Neural Information Processing Systems (NeurIPS 2024).

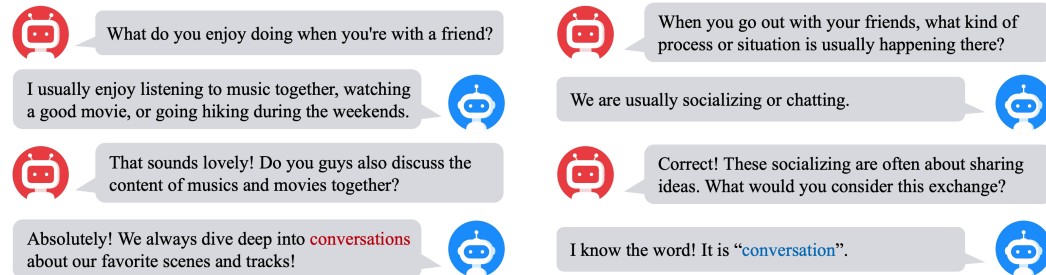

Figure 2: Examples of *Adversarial Taboo* with the same target word "conversation". The left shows an attacker-winning game, in which the defender unconsciously speaks out the target word. The right is a defender-winning episode because the defender makes the correct inference from the dialogue.

natural language understanding [Yang et al., 2023b, Touvron et al., 2023], text generation [Kocoń et al., 2023, Anil et al., 2023], machine translation [Jiao et al., 2023], and programming [Surameery and Shakor, 2023, Tian et al., 2023]. However, the reasoning ability of LLMs, which is essential for complex problem-solving [Pan et al., 2023] and advanced intelligence-developing [Yao et al., 2021], still retains being challenged by various criteria including correctness [Zhang et al., 2023a] and faithfulness [Turpin et al., 2024].

To address the reasoning challenge of LLMs, plenty of works have contributed in-depth efforts from the perspectives of Chain-of-Thought (CoT) prompt engineering [Wei et al., 2022, Ding et al., 2023, Yao et al., 2024], and the usage of auxiliary reasoning tools [Pan et al., 2023]. However, both prompt-based and tool-calling methods require additional prompt designs, which are inconsistent and sensitive to different prompt patterns and LLM checkpoints [Turpin et al., 2024, Chu et al., 2023]. More fundamental and consistent reasoning-improving approaches are post-pretraining [Azerbayev et al., 2023] and fine-tuning [Dong et al., 2023a], which trains LLMs with additional reasoning-related text corpus. Nevertheless, these methods demand sufficient high-quality textual data, which are difficult to collect due to the massive costs of human annotation efforts [Singh et al., 2023].

To improve LLM reasoning more efficiently, self-improvement methods, which enhance LLMs with model-generated synthetic data, have recently attracted increasing research attention [Singh et al., 2023, Huang et al., 2023, Burns et al., 2023, Chen et al., 2024]. Self-improvement methods usually utilize the intrinsic language capability of LLMs to judge [Huang et al., 2023], filter [Yuan et al., 2024], or revise [Yuan et al., 2024] self-generated samples to enhance their quality. However, most self-improvement methods rely on a broad range of high-quality question queries to prevent over-fitting into a sub-domain of reasoning tasks, which still requires additional data collection and cleaning. Besides, the judgments from LLMs are not guaranteed objective [Raina et al., 2024]. If an LLM already has an incorrect or biased recognition of a particular concept, the self-improvement process can reinforce and amplify the LLM's cognitive dissonance.

Towards more general and objective self-reasoning-improvement methods, we are inspired by the advancement from AlphaGO [Silver et al., 2016] to AlphaGO Zero [Silver et al., 2017], in which the game agents' intelligence continuously promotes via self-play without any human knowledge. Analogically, we expect to set up a language game where LLMs can improve their reasoning capacities via reinforcement learning (RL) during self-play. Although language games have attracted increasing attention in natural language processing [Lewis et al., 2017, Hausknecht et al., 2020, Xu et al., 2023, Wu et al., 2024], most of them are specially designed with customized game rules, in lack of the generalization to improve the general language capacities of LLMs. Among a few general-target language games including red-teaming [Ma et al., 2023], negotiation [Lewis et al., 2017], and bargain [Abdulhai et al., 2023], additional human judgments or reward models are required for outcome determination, which posts challenges on the efficiency and effectiveness of large-scale self-play RL training. Recent studies have raised interest in entity- or word-based language games, such as *20-Question* [Zhang et al., 2023b] and *Guess-My-City* [Abdulhai et al., 2023], which provide not only straight-forward word-level outcomes but also language universality by traversing the game word from comprehensive vocabularies. However, unlike the GO game, these word-based games are out of the adversarial scheme, limiting the game intensity and self-play learning effectiveness.

With the above consideration, we select an adversarial language game called *Adversarial Taboo* [Yao et al., 2021], in which an attacker and a defender perform a conversation around a target word

only visible to the attacker. The attacker aims to induce the defender to speak out the target word unconsciously; the defender tries to avoid unconscious utterance of the target word and guess the word from the dialogue history. To win the adversarial game in information-limited conversations, both players are required to have high-level language capacities in terms of expression, upstanding, and reasoning. Moreover, by collecting target words from a vast vocabulary, this game can cover a broad range of topics providing sufficient language versatility. Besides, the game outcomes can be automatically and explicitly judged: we only need to check whether the target word appears in the defender's utterances (attacker wins) or its inference patterns (defender wins). We conduct the self-play on this adversarial game using open-source LLMs, LLaMA-2-7B [Touvron et al., 2023] and Baichuan-2-13B [Yang et al., 2023a], with target words selected from a 50K top-frequency vocabulary [Davies, 2020]. Next, we conduct offline reinforcement learning on the game outcomes and observe significant performance improvement on a broad range of LLM reasoning benchmarks. Furthermore, we iterate this sampling-learning process with three epochs, within which the LLMs' reasoning can continuously obtain improvement. We believe this novel training scheme, **S**elf-**P**lay of **A**dversarial **G**ame (SPAG), has great potential for developing advanced LLM capacities.

## 2 Preliminary

With the development of LLMs [OpenAI, 2023a,b], reinforcement learning (RL) has played an increasingly important role in language model training [Ouyang et al., 2022, Ramamurthy et al., 2023]. The prime application scenario for RL in LLM training is reinforcement learning from human feedback (RLHF) [Yuan et al., 2023, Cheng et al., 2023b, Zeng et al., 2023]. RLHF first learns a reward model $r(\boldsymbol{x}, \boldsymbol{y})$ from the human feedback preference pairs [Cheng et al., 2023a], and then optimizes the LLM policy $\pi_\theta(\boldsymbol{y}|\boldsymbol{x})$ to maximize the expected reward value [Dong et al., 2023b]:

$$\mathcal{L}_{\text{RLHF}}(\pi_\theta) = -\mathbb{E}_{\boldsymbol{x}\sim\mathcal{D}, \boldsymbol{y}\sim\pi_\theta(\boldsymbol{y}|\boldsymbol{x})}[r(\boldsymbol{x}, \boldsymbol{y})]. \tag{1}$$

To learn the above objective, proximal policy optimization (PPO) [Schulman et al., 2017] algorithm has been recognized as the mainstream solution. In each update to equation 1, PPO minimizes:

$$\mathcal{L}_{\text{PPO}}(\pi_\theta) = -\mathbb{E}_{\boldsymbol{x}\sim\mathcal{D}, \boldsymbol{y}\sim\pi_{\bar{\theta}}(\boldsymbol{y}|\boldsymbol{x})}\left[\frac{\pi_\theta(\boldsymbol{y}|\boldsymbol{x})}{\pi_{\bar{\theta}}(\boldsymbol{y}|\boldsymbol{x})}\hat{A}^{\pi_{\bar{\theta}}} - \beta\text{KL}[\pi_{\bar{\theta}}\|\pi_\theta]\right], \tag{2}$$

where $\pi_{\bar{\theta}}$ is a copy of $\pi_\theta$ before the update, $\hat{A}^{\pi_{\bar{\theta}}}$ is the estimated advantage value [Schulman et al., 2016] with respect to the reference policy $\pi_{\bar{\theta}}$, and $\text{KL}[\pi_{\bar{\theta}}\|\pi_\theta]$ is the Kullback-Leibler (KL) [Kullback, 1997] divergence regularizing $\pi_\theta$ with an appropriate updating step. However, PPO for LLMs has been continually challenged due to its inefficient natural-text online sampling and the unstable training processes [Baheti et al., 2024]. Among the improvements to PPO [Rafailov et al., 2023, Yuan et al., 2023, Dong et al., 2023b], Baheti et al. [2024] adopts the PPO objective into an offline scheme by using importance sampling [Neal, 2001], which is named Advantage-Leftover-Lunch (A-LoL):

$$\nabla_\theta\hat{\mathcal{L}}_{\text{RLHF-A-LoL}} = \mathbb{E}_{\boldsymbol{x}\sim\mathcal{D}, \boldsymbol{y}\sim\pi_{\text{ref}}(\boldsymbol{y}|\boldsymbol{x})}\left[\hat{A}^{\pi_{\text{ref}}} \cdot \frac{\pi_\theta(\boldsymbol{y}|\boldsymbol{x})}{\pi_{\text{ref}}(\boldsymbol{y}|\boldsymbol{x})} \cdot \nabla_\theta\log\pi_\theta(\boldsymbol{y}|\boldsymbol{x})\right]. \tag{3}$$

Here the sample $\boldsymbol{y} \sim \pi_{\text{ref}}(\boldsymbol{y}|\boldsymbol{x})$ and advantage $\hat{A}^{\pi_{\text{ref}}}$ are both from the reference distribution $\pi_{\text{ref}}(\boldsymbol{y}|\boldsymbol{x})$ and calculated offline. Besides, Gulcehre et al. [2023] proposed Reinforced Self-Training (ReST) to simplify the RLHF scheme. With a threshold $\tau \in \mathbb{R}$, ReST updates the LLM by the reinforcement on the selected samples $\mathcal{D}_\tau = \{(\boldsymbol{x}, \boldsymbol{y}) : r(\boldsymbol{x}, \boldsymbol{y}) > \tau\}$:

$$\mathcal{L}_{\text{ReST}}(\pi_\theta) = \mathbb{E}_{\boldsymbol{x}\sim\mathcal{D}, \boldsymbol{y}\sim\pi_{\text{ref}}(\boldsymbol{y}|\boldsymbol{x})}\left[\mathbf{1}_{r(\boldsymbol{x},\boldsymbol{y})>\tau} \cdot \mathcal{L}_\theta(\boldsymbol{x}, \boldsymbol{y})\right] = \mathbb{E}_{\mathcal{D}_\tau}[\mathcal{L}_\theta(\boldsymbol{x}, \boldsymbol{y})], \tag{4}$$

where $\mathcal{L}_\theta(\boldsymbol{x}, \boldsymbol{y})$ could be any offline RL loss such as A-LoL or the vanilla language modeling loss.

## 3 Self-play of Adversarial Language Games

The game of Adversarial Taboo is first introduced by Yao et al. [2021], in which an attacker $\mu$ and a defender $\nu$ involve a multi-turn conversation. At the beginning of the game, the attacker is assigned a target word $w \in \mathcal{V}_{\text{target}}$, which is not informed to the defender. The attacker's target is to induce the defender to speak the target word unconsciously. To achieve this goal, the attacker can talk about any topic related to $w$, except directly speaking out the target word. In contrast, the defender is required to infer the target word without any unconscious utterance of the word. If the defender has sufficient confidence to infer the word, it can yell "I know the word! It is {*target word*}!". Then the game terminates. If the guess is correct, the defender wins, otherwise the attacker wins. Besides, the game has a maximum number of turns $T_0$. If nobody wins during $T_0$ turns, there is a tie. The examples of Adversarial Taboo are shown in Figure 2.

---

**Algorithm 1** Data collection of LLM self-plays for the adversarial language game.

---

**Inputs:** LLM policy $\pi_\theta(\boldsymbol{y}|\boldsymbol{x})$, target word $w$, attacker and defender prompts $f_{\text{attack}}, f_{\text{defend}}$.
Set the initial state $\boldsymbol{s}_0 = (w)$.
**for** $t$ from 1 to $T$ **do**
    Sample an attacker utterance $\boldsymbol{u}_t \sim \mu_\theta(\boldsymbol{u}_t|\boldsymbol{s}_{t-1}) = \pi_\theta(\boldsymbol{y} = \boldsymbol{u}_t|\boldsymbol{x} = f_{\text{attack}}(\boldsymbol{s}_{t-1}))$.
    Update state $\boldsymbol{s}'_t = (w, \boldsymbol{u}_1, \boldsymbol{v}_1, \ldots, \boldsymbol{u}_{t-1}, \boldsymbol{v}_{t-1}, \boldsymbol{u}_t)$.
    Sample a defender utterance $\boldsymbol{v}_t \sim \nu_\theta(\boldsymbol{v}_t|\boldsymbol{s}'_t) = \pi_\theta(\boldsymbol{y} = \boldsymbol{v}_t|\boldsymbol{x} = f_{\text{defend}}(\boldsymbol{s}'_t))$.
    Update state $\boldsymbol{s}_t = (w, \boldsymbol{u}_1, \boldsymbol{v}_1, \ldots, \boldsymbol{u}_{t-1}, \boldsymbol{v}_{t-1}, \boldsymbol{u}_t, \boldsymbol{v}_t)$.
**end for**
Collect an episode $\tau = (\boldsymbol{s}_0, \boldsymbol{s}'_1, \boldsymbol{s}_1, \ldots, \boldsymbol{s}'_T, \boldsymbol{s}_T)$

---

## 3.1 Adversarial Language Game Modeling

We view the Adversarial Taboo as a two-player zero-sum Markov game [Littman, 1994], which can be described by a tuple as $(\mathcal{S}, \mathcal{A}, F, r)$:

- The state space $\mathcal{S} = \{\boldsymbol{s}_t, \boldsymbol{s}'_t : 1 \leq t \leq T_0\}$ contains two types of states, $\boldsymbol{s}'_t = (w, \boldsymbol{u}_1, \boldsymbol{v}_1, \boldsymbol{u}_2, \ldots, \boldsymbol{u}_t)$ and $\boldsymbol{s}_t = (w, \boldsymbol{u}_1, \boldsymbol{v}_1, \boldsymbol{u}_2, \ldots, \boldsymbol{u}_t, \boldsymbol{v}_t)$, where $\{\boldsymbol{u}_i\}_{i=1}^t$ and $\{\boldsymbol{v}_i\}_{i=1}^t$ are the utterances of the attacker and defender, respectively. Games start at $\boldsymbol{s}_0 = (w)$ with a target word $w \in \mathcal{V}_{\text{target}}$ and end with $T_0$ maximum turns. States $\boldsymbol{s}'_t$ and $\boldsymbol{s}_t$ end with utterances $\boldsymbol{u}_t$ and $\boldsymbol{v}_t$ for the defender and attacker to act, respectively.

- The action space $\mathcal{A}$ is shared with both the attacker and defender, which is equivalent to the token sequence space of natural language $\mathcal{N} = \{\boldsymbol{x} = (x_1, x_2, \ldots, x_L)|x_l \in \mathcal{V}_{\text{token}}, L \in \mathbb{N}_+\}$.

- The transition function $F : \mathcal{S} \times \mathcal{A} \to \mathcal{S}$ deterministically appends the utterance $\boldsymbol{u}_t$ or $\boldsymbol{v}_t$ at the end of the dialogue, and converts $\boldsymbol{s}'_t = F(\boldsymbol{s}_{t-1}, \boldsymbol{u}_t)$ and $\boldsymbol{s}_t = F(\boldsymbol{s}'_t, \boldsymbol{v}_t)$.

- The reward $r : \mathcal{S} \times \mathcal{A} \to \mathbb{R}$ evaluates the actions $\boldsymbol{u}, \boldsymbol{v} \in \mathcal{A}$ based on their corresponding states $\boldsymbol{s}, \boldsymbol{s}' \in \mathcal{S}$ with rewards $r(\boldsymbol{s}, \boldsymbol{u})$ and $r(\boldsymbol{s}', \boldsymbol{v})$, respectively. Given a game episode $\tau = (\boldsymbol{s}_0, \boldsymbol{s}'_1, \boldsymbol{s}_1, \ldots, \boldsymbol{s}'_T, \boldsymbol{s}_T)$, we denote the *attacker's total reward $R(\tau) = \sum_{t=1}^T r(\boldsymbol{s}_{t-1}, \boldsymbol{u}_t)$*, so the *defender's total reward* is $\sum_{t=1}^T r(\boldsymbol{s}'_t, \boldsymbol{v}_t) = -R(\tau)$ to satisfy the *zero-sum* constraint. More detailed reward designs with heuristic rules for the Adversarial Taboo can be found in Appendix B.

In the above game, we denote $\mu(\boldsymbol{u}|\boldsymbol{s})$ and $\nu(\boldsymbol{v}|\boldsymbol{s}')$ as the attacker's and defender's policies, respectively. Then each episode $\tau$ can be regarded as a trajectory with the probability:

$$P(\tau) = P(\boldsymbol{s}_0) \prod_{t=1}^T P(\boldsymbol{s}'_t|\boldsymbol{s}_{t-1}) \prod_{t=1}^T P(\boldsymbol{s}_t|\boldsymbol{s}'_t) = P(w) \prod_{t=1}^T \mu(\boldsymbol{u}_t|\boldsymbol{s}_{t-1}) \prod_{t=1}^T \nu(\boldsymbol{v}_t|\boldsymbol{s}'_t) =: (\mu \times \nu), \quad (5)$$

where $P(w)$ is the data distribution of target word $w \in \mathcal{V}_{\text{target}}$. Then we can write the self-play objective of the Adversarial Taboo as:

$$\max_\mu \min_\nu \mathcal{L}_{\text{AG}}(\mu, \nu) := \mathbb{E}_{\tau \sim \mu \times \nu}[R(\tau)], \quad (6)$$

in which the attacker tries to maximize its total reward $R(\tau)$ by optimizing policy $\mu$, and the defender seeks strategies $\nu$ to maximize the defender reward $-R(\tau)$ (minimize $R(\tau)$). To play the above adversarial game with an LLM generation policy $\pi_\theta(\boldsymbol{y}|\boldsymbol{x})$, we first design prompt templates $f_{\text{attack}}, f_{\text{defend}} : \mathcal{S} \to \mathcal{N}$ for the attacker and defender respectively to convert the game states into natural language task descriptions. Next, we introduce the game policies for the two players:

$$\mu_\theta(\boldsymbol{u}|\boldsymbol{s}) = \pi_\theta(\boldsymbol{u}|f_{\text{attack}}(\boldsymbol{s})), \text{ and } \nu_\theta(\boldsymbol{v}|\boldsymbol{s}') = \pi_\theta(\boldsymbol{v}|f_{\text{defend}}(\boldsymbol{s}')). \quad (7)$$

The detailed prompt templates for the game are demonstrated in Appendix A.

## 3.2 Imitation Learning

Due to the limited capability of current open-source LLMs, the generation policy $\pi_\theta(\boldsymbol{y}|\boldsymbol{x})$ can not guarantee the strict instruction-following of the game rules in prompts $f_{\text{attack}}(\boldsymbol{s})$ and $f_{\text{defend}}(\boldsymbol{s}')$. Therefore, before the self-play, we first conduct an imitation learning (behavior cloning) of GPT-4's behaviors to ensure that $\pi_\theta(\boldsymbol{u}|f_{\text{attack}}(\boldsymbol{s}))$ and $\pi_\theta(\boldsymbol{v}|f_{\text{defend}}(\boldsymbol{s}'))$ act consistently with the game rules. To collect the game episodes of GPT-4 [Achiam et al., 2023], we use the data collection procedure

described in Algorithm 1. Similar to the setups in equation 7, we also design attacker and defender prompts for GPT-4 to act as the game players, which can be found in Appendix A.1.

After collecting a group of GPT-4 game episodes for imitation learning as $\mathcal{T}_{\text{im}}$, we divide it into an attacker-winning set $\mathcal{T}_{\text{im}}^{\text{attack}} = \{\boldsymbol{\tau} \in \mathcal{T}_{\text{im}} : R(\boldsymbol{\tau}) > 0\}$ and a defender-winning set $\mathcal{T}_{\text{im}}^{\text{defend}} = \{\boldsymbol{\tau} \in \mathcal{T}_{\text{im}} : R(\boldsymbol{\tau}) < 0\}$. The imitation learning loss is to maximize the log-likelihood of winners' actions:

$$\mathcal{L}_{\text{im}}^{\text{attack}}(\pi_\theta) = - \mathbb{E}_{\boldsymbol{\tau} \in \mathcal{T}_{\text{im}}^{\text{attack}}} \Big[ \frac{1}{T} \sum_{t=1}^{T} \log \pi_\theta(\boldsymbol{u}_t | f_{\text{attack}}(\boldsymbol{s}_{t-1})) + \beta_1 \text{KL}[\pi_\theta \| \pi_{\text{ref}}] \Big], \tag{8}$$

$$\mathcal{L}_{\text{im}}^{\text{defend}}(\pi_\theta) = - \mathbb{E}_{\boldsymbol{\tau} \in \mathcal{T}_{\text{im}}^{\text{defend}}} \Big[ \frac{1}{T} \sum_{t=1}^{T} \log \pi_\theta(\boldsymbol{v}_t | f_{\text{defend}}(\boldsymbol{s}'_t)) + \beta_1 \text{KL}[\pi_\theta \| \pi_{\text{ref}}] \Big], \tag{9}$$

where the re-weighting parameter $\beta_1 > 0$ and the regularizer $\text{KL}[\pi_\theta \| \pi_{\text{ref}}]$ prevents the model from over-fitting on the language game task and maintains the model's general language abilities. The reference model $\pi_{\text{ref}}$ is the initial checkpoint of the LLM before training. The overall imitation learning objective is $\mathcal{L}_{\text{im}}(\pi_\theta) = \frac{1}{2}\mathcal{L}_{\text{im}}^{\text{attack}}(\pi_\theta) + \frac{1}{2}\mathcal{L}_{\text{im}}^{\text{defend}}(\pi_\theta)$.

### 3.3 Reinforcement Learning from Self-play

Imitation learning enables LLM to behave consistently following the game rules. Next, we conduct self-play by letting the LLM $\pi_\theta(\boldsymbol{y}|\boldsymbol{x})$ play alternatively as the attacker $\mu_\theta(\boldsymbol{u}|\boldsymbol{s}) = \pi_\theta(\boldsymbol{u}|f_{\text{attack}}(\boldsymbol{s}))$ and the defender $\nu_\theta(\boldsymbol{v}|\boldsymbol{s}') = \pi_\theta(\boldsymbol{v}|f_{\text{defend}}(\boldsymbol{s}'))$. Note that the self-play sampling process involves massive multi-turn auto-regressive text generation of the LLM, which causes heavy computational complexity and makes the on-policy RL training intensively inefficient. Therefore, we use an offline learning scheme: (1) make a copy $\pi_{\bar{\theta}}$ of current LLM policy $\pi_\theta$; (2) collect self-play episodes $\mathcal{T}_{\bar{\theta}} = \{\boldsymbol{\tau} \sim \mu_{\bar{\theta}} \times \nu_{\bar{\theta}}\}$ from games between attacker $\mu_{\bar{\theta}}$ and defender $\nu_{\bar{\theta}}$; (3) update $\pi_\theta$ via RL training with $\mathcal{T}_{\bar{\theta}}$. The details of the collection of $\mathcal{T}_{\bar{\theta}}$ are shown in Algorithm 1.

With a group of episodes $\mathcal{T}_{\bar{\theta}}$, we first fix the defender policy $\nu_{\bar{\theta}}$ and consider updating the attacker policy $\mu_\theta$ with respect to the self-play objective $\mathcal{L}_{\text{AG}}(\mu_\theta, \nu_{\bar{\theta}}) = \mathbb{E}_{\boldsymbol{\tau} \sim \mu_\theta \times \nu_{\bar{\theta}}}[R(\boldsymbol{\tau})]$. We calculate the corresponding policy gradient for the attacker as:

$$\nabla_\theta \mathcal{L}_{\text{AG}}(\mu_\theta, \nu_{\bar{\theta}}) = \mathbb{E}_{\boldsymbol{\tau} \sim \mu_\theta \times \nu_{\bar{\theta}}} \Big[ \sum_{t=1}^{T} A_t^{\mu_\theta} \cdot \nabla_\theta \log \mu_\theta(\boldsymbol{u}_t | \boldsymbol{s}_{t-1}) \Big]$$

$$= \mathbb{E}_{\boldsymbol{\tau} \sim \mu_{\bar{\theta}} \times \nu_{\bar{\theta}}} \Big[ \sum_{t=1}^{T} A_t^{\mu_\theta} \cdot \frac{\mu_\theta(\boldsymbol{u}_t | \boldsymbol{s}_{t-1})}{\mu_{\bar{\theta}}(\boldsymbol{u}_t | \boldsymbol{s}_{t-1})} \cdot \nabla_\theta \log \mu_\theta(\boldsymbol{u}_t | \boldsymbol{s}_{t-1}) \Big], \tag{10}$$

where $A_t^{\mu_\theta} = A^{\mu_\theta}(\boldsymbol{s}_{t-1}, \boldsymbol{u}_t)$ is the advantage of action $\boldsymbol{u}_t$ for the attacker $\mu_\theta$ in the self-play of $\mu_\theta \times \nu_{\bar{\theta}}$. Here we apply importance sampling to unbiasedly estimate the expectation *w.r.t* $\mu_\theta(\boldsymbol{u}_t | \boldsymbol{s}_{t-1})$ with the sampled actions from $\mu_{\bar{\theta}}(\boldsymbol{u}_t | \boldsymbol{s}_{t-1})$ in $\mathcal{T}_{\bar{\theta}}$. Inspired by TRPO [Schulman et al., 2015] and PPO [Schulman et al., 2017], we design the following loss to optimize $\mathcal{L}_{\text{AG}}(\mu_\theta, \nu_{\bar{\theta}})$:

$$\mathcal{L}_{\text{sp}}^{\text{attack}}(\pi_\theta) = - \mathbb{E}_{\boldsymbol{\tau} \in \mathcal{T}_{\bar{\theta}}} \Big[ \sum_{t=1}^{T} \frac{\mu_\theta(\boldsymbol{u}_t | \boldsymbol{s}_{t-1})}{\mu_{\bar{\theta}}(\boldsymbol{u}_t | \boldsymbol{s}_{t-1})} \hat{A}_t^{\mu_{\bar{\theta}}} - \beta_2 \text{KL}[\pi_\theta \| \pi_{\bar{\theta}}] \Big]$$

$$= - \mathbb{E}_{\boldsymbol{\tau} \in \mathcal{T}_{\bar{\theta}}} \Big[ \sum_{t=1}^{T} \frac{\pi_\theta(\boldsymbol{u}_t | f_{\text{attack}}(\boldsymbol{s}_{t-1}))}{\pi_{\bar{\theta}}(\boldsymbol{u}_t | f_{\text{attack}}(\boldsymbol{s}_{t-1}))} \hat{A}_t^{\mu_{\bar{\theta}}} - \beta_2 \text{KL}[\pi_\theta \| \pi_{\bar{\theta}}] \Big], \tag{11}$$

where the re-weighting parameter $\beta_2 > 0$, and regularizer $\text{KL}(\pi_\theta \| \pi_{\bar{\theta}})$ guarantees an appropriate policy update step size. Following TRPO [Schulman et al., 2015], we use empirical estimation $\hat{A}_t^{\mu_{\bar{\theta}}}$ of policy $\mu_{\bar{\theta}}$ to approximate the advantage $A_t^{\mu_\theta}$. More details of advantage estimation are described in Appendix D. Similarly, from the perspective of the defender, we propose the corresponding loss:

$$\mathcal{L}_{\text{sp}}^{\text{defend}}(\pi_\theta) = - \mathbb{E}_{\boldsymbol{\tau} \in \mathcal{T}_{\bar{\theta}}} \Big[ \sum_{t=1}^{T} \frac{\pi_\theta(\boldsymbol{v}_t | f_{\text{defend}}(\boldsymbol{s}'_t))}{\pi_{\bar{\theta}}(\boldsymbol{v}_t | f_{\text{defend}}(\boldsymbol{s}'_t))} \hat{A}_t^{\nu_{\bar{\theta}}} - \beta_2 \text{KL}[\pi_\theta \| \pi_{\bar{\theta}}] \Big]. \tag{12}$$

In practice, we find when negative advantage values exist, the above policy gradient method can cause training instability and damage the LLMs' general language performance. To mitigate the

**Algorithm 2** Self-play of adversarial language games (SPAG)

---

**Inputs:** LLM policy $\pi_\theta(\boldsymbol{y}|\boldsymbol{x})$, target word set $\mathcal{V}_{\text{target}}$, attacker and defender prompts $f_{\text{attack}}, f_{\text{defend}}$.
**for** self-play epoch iterations **do**
    Make a copy of $\pi_\theta$ as $\pi_{\bar{\theta}}$.
    Set $\mu_{\bar{\theta}}(\boldsymbol{u}|\boldsymbol{s}) = \pi_{\bar{\theta}}(\boldsymbol{u}|f_{\text{attack}}(\boldsymbol{s}))$ and $\nu_{\bar{\theta}}(\boldsymbol{v}|\boldsymbol{s}') = \pi_{\bar{\theta}}(\boldsymbol{v}|f_{\text{defend}}(\boldsymbol{s}'))$.
    For each $w \in \mathcal{V}_{\text{target}}$, sample a episode $\boldsymbol{\tau} \sim \mu_{\bar{\theta}} \times \nu_{\bar{\theta}}$. Collect $\mathcal{T}_{\bar{\theta}} = \{\boldsymbol{\tau} \sim \mu_{\bar{\theta}} \times \nu_{\bar{\theta}}\}$
    Split the attacker-winning set $\mathcal{T}_{\bar{\theta}}^{\text{attack}}$ and the defender-winning set $\mathcal{T}_{\bar{\theta}}^{\text{defend}}$.
    Update $\pi_\theta$ with loss $\mathcal{L}_{\text{SPAG}}(\pi_\theta)$.
**end for**

---

issue and obtain more stable RL training, we utilize the methodology of ReST [Gulcehre et al., 2023] as in equation 4, which considers the offline learning with samples $\{\boldsymbol{\tau} : R(\boldsymbol{\tau}) > \xi\}$ selected by a reward threshold $\xi$. More specifically, we set the reward threshold $\xi = 0$ and select the attacker-winning episodes $\mathcal{T}_{\bar{\theta}}^{\text{attack}} = \{\boldsymbol{\tau} \in \mathcal{T}_{\bar{\theta}} : R(\boldsymbol{\tau}) > 0\}$ and the defender-winning episodes $\mathcal{T}_{\bar{\theta}}^{\text{defend}} = \{\boldsymbol{\tau} \in \mathcal{T}_{\bar{\theta}} : R(\boldsymbol{\tau}) < 0\}$ for the attacker and the defender training respectively. Similar techniques have also been studied in earlier RL literature to obtain stable policy updates, such as self-imitation learning [Oh et al., 2018] and the UPGO [Vinyals et al., 2019] methods. Therefore, the overall self-play of adversarial language games (SPAG) objective is:

$$\mathcal{L}_{\text{SPAG}}(\pi_\theta) = -\frac{1}{2}\mathbb{E}_{\mathcal{T}_{\bar{\theta}}^{\text{attack}}}\Big[\sum_{t=1}^{T}\frac{\pi_\theta(\boldsymbol{u}_t|f_{\text{attack}}(\boldsymbol{s}_{t-1}))}{\pi_{\bar{\theta}}(\boldsymbol{u}_t|f_{\text{attack}}(\boldsymbol{s}_{t-1}))}\hat{A}_t^{\mu_{\bar{\theta}}} - \beta_2\text{KL}[\pi_\theta\|\pi_{\bar{\theta}}]\Big] \tag{13}$$

$$-\frac{1}{2}\mathbb{E}_{\mathcal{T}_{\bar{\theta}}^{\text{defend}}}\Big[\sum_{t=1}^{T}\frac{\pi_\theta(\boldsymbol{v}_t|f_{\text{defend}}(\boldsymbol{s}'_t))}{\pi_{\bar{\theta}}(\boldsymbol{v}_t|f_{\text{defend}}(\boldsymbol{s}'_t))}\hat{A}_t^{\nu_{\bar{\theta}}} - \beta_2\text{KL}[\pi_\theta\|\pi_{\bar{\theta}}]\Big] - \alpha\mathbb{E}_{(\boldsymbol{x},\boldsymbol{y})\sim\mathcal{D}_{\text{SFT}}}[\log\pi_\theta(\boldsymbol{y}|\boldsymbol{x})],$$

where $\alpha > 0$ is a re-weighting hyper-parameter, and $\mathbb{E}_{\mathcal{D}_{\text{SFT}}}[\log\pi_\theta(\boldsymbol{y}|\boldsymbol{x})]$ is the log-likelihood on a supervised fine-tuning (SFT) dataset $\mathcal{D}_{\text{SFT}}$ to prevent LLMs from losing general language abilities.

## 4 Experiments

To verify the effectiveness of SPAG, we select open-source pretrained LLMs of different sources and model sizes, particularly LLaMA-2-7B [Touvron et al., 2023] and Baichuan-2-13B [Yang et al., 2023a]. As introduced in Section 3, the training process includes two stages: imitation learning of GPT-4, and self-play learning on game episodes. For baseline comparison, we consider Chain-of-Thought (CoT) [Wei et al., 2022] and continuous supervised fine-tuning (SFT) methods. Besides, we also test another two keyword-based non-adversarial language games: *20-Question* and *Guess-My-City* as described in Abdulhai et al. [2023]. More details about the two games are in Appendix E.

### 4.1 Experimental Setups

**Training Data Preparation**    The training data preparation consists of the following three parts. More data collection details are in Appendix A.

- *Target Words:*  We aim to play the adversarial game with an extensive range of target words so that diverse topics can be discussed during the self-play processes, which helps maintain the generalization ability of LLMs. Hence, we collect the 50K most frequently used words from the Corpus of Contemporary American (CoCA) [Davies, 2020] as the target word list $\mathcal{V}_{\text{target}}$. Besides, stop words defined in NLTK [Bird, 2006], which are commonly used with insignificant semantic meaning, are filtered out of $\mathcal{V}_{\text{target}}$.

- *Imitation Learning Data:*  To enable the instruction-following ability of open-source LLMs on game rules, we use the same data collection process in Algorithm 1 via the GPT-4 [OpenAI, 2023b] API and play the Taboo game one episode per target word. The attacker and defender prompts are in Appendix A.1. Due to the resource limitation, we only collect the GPT-4 self-play samples with the top 30K words from $\mathcal{V}_{\text{target}}$. The maximum interaction turn is randomly selected in the range $[3, 8]$. The results are collected as $\mathcal{T}_{\text{im}}$ for the imitation learning.

- *Supervised Funetuning (SFT) Data:*  We also prepare general query-response SFT data to prevent LLMs from being over-fitted on the adversarial game. We use Alpaca [Taori et al., 2023] as the SFT set, which contains 52K instruction-following data from GPT-3 [Brown et al., 2020].

Table 1: Reasoning Performance of SPAG on LLaMA-2-7B.

| | MMLU | BBH | Mutual | ARC-e | ARC-c | LGQA2 | WGrande | PIQA | GM (Avg.) |
|---|---|---|---|---|---|---|---|---|---|
| LLaMA-2-7B | 45.80 | 32.48 | 50.90 | 76.30 | 43.26 | 25.32 | 69.14 | 78.07 | 49.17 |
| LLaMA-2-7B-CoT | 44.62 | **38.73*** | 52.03 | 73.44 | 40.96 | 25.89 | **71.82*** | 78.35 | 50.05 |
| AlpacaSFT-1 | 35.17 | 30.24 | 53.95 | 76.81 | 44.97 | 28.94 | 69.61 | 78.07 | 48.61 |
| AlpacaSFT-2 | 44.17 | 32.50 | 55.08 | 77.15 | 46.50 | 29.20 | 68.67 | 78.24 | 50.82 |
| AlpacaSFT-3 | 45.87 | 31.52 | 54.18 | 75.25 | 45.05 | 29.07 | 66.85 | 76.71 | 50.08 |
| AlpacaSFT-3-CoT | 44.70 | 34.56 | 54.18 | 74.37 | 42.32 | 29.13 | 67.72 | 76.55 | 50.11 |
| Imitation-*20Q* | 36.93 | 29.61 | 49.89 | 73.48 | 39.33 | 25.70 | 69.22 | 76.93 | 46.43 |
| Imitation-*GuessCity* | 46.13 | 32.82 | 51.58 | 76.22 | 43.09 | 25.95 | 68.82 | 78.13 | 49.46 |
| Imitation-AG | 46.15 | 32.74 | 52.82 | 76.81 | 44.80 | 27.10 | 69.46 | 78.24 | 50.22 |
| SP-*20Q* | 37.91 | 30.58 | 51.35 | 75.46 | 42.32 | 26.78 | 69.30 | 77.37 | 47.79 |
| SP-*GuessCity* | 45.32 | 31.64 | 50.56 | 75.34 | 42.15 | 25.57 | 69.22 | 78.51 | 48.78 |
| IM-AlpacaSFT | 46.50 | 34.03 | 54.18 | 76.86 | 45.55 | 29.20 | 68.82 | 78.31 | 51.20 |
| SPAG-1 | 47.01 | 34.39 | 54.85 | 77.69 | 45.65 | 29.83 | 68.90 | 78.89 | 51.69 |
| SPAG-2 | **47.28** | 34.73 | 54.97 | 78.45 | 46.84 | 30.08 | 69.61 | 79.33 | 52.19 |
| SPAG-3 | 47.11 | **34.94** | **55.30** | **78.54** | **47.53** | **30.98** | 69.93 | **79.38** | **52.58** |

**Evaluation**  To evaluate the LLMs' performance, we consider the following two perspectives:

- *Reasoning Benchmarks:*  To test the reasoning ability, we consider the following commonly used benchmarks including BIG-Bench Hard (BBH) [Suzgun et al., 2022], ARC easy (ARC-e) & challenge (ARC-c) [Clark et al., 2018], Mutual [Cui et al., 2020], WinoGrande [Sakaguchi et al., 2019], LogiQA2 [Liu et al., 2023], PIQA [Bisk et al., 2020]. BBH requires the exact match of the generated answers, the other benchmarks are all within the multiple-choice form. Besides reasoning metrics, we test MMLU [Hendrycks et al., 2020] as a general performance evaluation for LLMs. Additionally, we calculate the geometric mean of the numerical results of all benchmarks as an overall performance measurement. More details can be found in Appendix C.

- *Game Win Rates:*  Besides the general LLM capabilities, we report the win rates on a testing vocabulary $\mathcal{V}_{\text{test}}$ to validate if the game skills of LLMs improve through self-play. Following Abdulhai et al. [2023], we use the same keyword list in *20-Question* as $\mathcal{V}_{\text{test}}$, which contains 168 typical words manually selected from diverse daily objects. We denote the number of winning episodes as $N_{\text{win}}$, the number of losing episodes as $N_{\text{lose}}$, and the number of tied games as $N_{\text{tie}}$. Then, the player win rate is calculated as $(N_{\text{win}} + 0.5N_{\text{tie}})/(N_{\text{win}} + N_{\text{loss}} + N_{\text{tie}})$. The invalid game episodes, where LLM players do not strictly follow the game rules, are ignored.

**Training Details**  Most of our LLM training setups follow Alpaca [Taori et al., 2023]. For imitation learning, the learning rate is 5e-6, and the KL-penalty coefficient $\beta_1 = 0.1$. For SPAG training, the learning rate is 2e-6, the KL-penalty coefficient $\beta_2 = 0.2$, and the SFT coefficient $\alpha = 0.5$. For the Alpaca SFT baseline, we exactly follow the training setups of Alpaca and set the learning rate to 2e-6. Among all training stages, the batch size is 128 and the max sequence length is 2048. Each training process maintains one epoch over the offline collected trajectories. The the decay parameter $\gamma$ is set to 0.8. The maximum turn of the Adversarial Taboo is 5. All our experiments are conducted using 32 NVIDIA A100-SXM4 GPUs with 40GB memory.

### 4.2  Results Analysis

The main results are shown in Figure 1, where each axis is normalized by the maximum performance value, representing the reasoning performance on a particular benchmark. For notation simplification, we call the LLM obtained after imitating learning as the IM model, and the LLM trained after the $i$-th epoch of SPAG as SPAG-$i$ ($i = 1, 2, 3$).

**Imitation Learning**  With imitation learning over the collected GPT-4 self-play episodes, both LLaMA-2-7B and Baichuan-2-13B have obtained uniform performance improvements on all the reasoning benchmarks. As shown in Figure 1, both gray regions are completely wrapped by the blue-line polygons. Besides, as for the general language capacity, the imitation-learned (IM) LLaMA-2 model achieves a better MMLU performance than the original LLaMA-2 base in Table 2. Although the imitation result of Baichuan-2-13B on MMLU slightly underperforms the base model, the numerical difference is insignificant compared to the reasoning gains, which indicates the imitation learning of the GPT-4 game behaviors can improve LLM reasoning while preserving the general language capacity. In Table 1, we also report the IM models on two keyword-based games within

Table 2: Reasoning Performance of SPAG on Baichuan-2-13B.

| | MMLU | BBH | Mutual | ARC-e | ARC-c | LGQA2 | WGrande | PIQA | GM (Avg.) |
|---|---|---|---|---|---|---|---|---|---|
| Baichuan-2-13B | **59.00** | 39.03 | 53.72 | 77.36 | 46.84 | 30.73 | 69.93 | 77.42 | 54.21 |
| AlpacaSFT-1 | 52.94 | 36.52 | **58.35** | 74.12 | 44.88 | 33.33 | 71.35 | 77.20 | 53.68 |
| AlpacaSFT-2 | 51.27 | 36.60 | 57.67 | 73.36 | 44.28 | 33.46 | 69.77 | 75.90 | 53.00 |
| AlpacaSFT-3 | 52.14 | 36.57 | 55.08 | 69.44 | 42.15 | 33.91 | 66.61 | 74.59 | 51.79 |
| Imitation-AG | 58.37 | 39.49 | 57.11 | 76.60 | 47.53 | 33.65 | 70.59 | 78.49 | 55.45 |
| IM+AlpacaSFT | 57.45 | 39.60 | 58.01 | 77.61 | 48.55 | 34.10 | 70.60 | 78.50 | 55.80 |
| SPAG-1 | 57.93 | 39.81 | 57.45 | 78.20 | 48.55 | 35.05 | 70.67 | 78.69 | 56.09 |
| SPAG-2 | 57.99 | 39.97 | 57.67 | **78.32** | 49.83 | 35.62 | 71.03 | 78.83 | 56.52 |
| SPAG-3 | 57.75 | **40.30** | 57.79 | 78.11 | **50.00** | **36.26** | **71.43** | **79.05** | **56.75** |

non-adversarial setups: Imitation-*20Q* (on *20-Question*) and Imitation-*GuessCity* (on *Guess-My-City*). From the results, we find the IM model on Adversarial Taboo outperforms models trained on non-adversarial games, highlighting the effectiveness of the adversarial game setups for reasoning improvement. Besides, we report the Chain-of-Thought (CoT) reasoning results of LLaMA-2 (Lama-2-Base-CoT) and Alpaca-2 (AlpacaSFT-3-CoT). Although the CoT method on LLaMA-2 reaches conspicuous performance on BBH and WinoGrande, the IM model can still surpass the CoT results in terms of overall reasoning performance (geometric mean).

**Self-play Training** After imitation learning, we conduct three epochs of SPAG training as described in Algorithm 2. As shown in Figure 1, on most of the reasoning benchmarks, both LLaMA-2-7B and Baichuan-2-13B have their performance steadily improved with the SPAG training epoch increasing. For LLaMA-2-7B, although the first-epoch SPAG model has a relatively lower performance than the imitation-learned (IM) model on WinoGrande, after an additional epoch of self-play iteration, SPAG-2 has achieved sufficient improvement to surpass the performance of the IM model. Considering the general language capability on MMLU, SPAG models can not guarantee continuous improvement, especially for Baichuan-2-13B whose performance slightly decays during the SPAG training. Compared to the improvements in the reasoning benchmarks, this language-understanding decline is still within an acceptable range, since the overall performance (GM score) maintains increasing significantly. For the baseline comparison, we report the continuous SFT on the IM models (IM+AlpacaSFT) and self-played models on non-adversarial games (SP-*20Q* and SP-*GuessCity*). On both LLaMA-2 and Baichuan-2, continuous SFT models have lower performance scores compared with the corresponding SPAG-1 models. For non-adversarial self-play, SP-*GuessCity* even performs worse than Imitation-*GuessCity*, which with a high probability is because the game *Guess-My-City* has a more narrow topic range, insufficient to comprehensively improve the general LLM capacity.

**Ablation Study** Since the SPAG training loss includes the SFT data, we conduct an ablation study to test whether the performance gains on reasoning benchmarks come from the SFT data or the SPAG method. More specifically, we follow the training setups in Alpaca [Taori et al., 2023] and conduct SFT on LLaMA-2-7B and Baichuan-2-13B with three epochs. The checkpoint after the $i$th-epoch training is denoted as AlpacaSFT-$i$ ($i = 1, 2, 3$) and tested on the same evaluation sets. The SFT models' performances are also reported in Figure 1 and Table 1 & 2. With the LLaMA base, our SPAG-3 model can uniformly defeat the SFT baselines with clear performance gaps on all benchmarks. With the Baichuan base, except the Mutual set, the SPAG models maintain noticeable advantages on other metrics. Considering the distinct surpass of overall performance, we claim that the major contribution to the reasoning improvements should be credited to the SPAG scheme.

Moreover, we test the sample efficiency and hyper-parameter effectiveness of SPAG in Figure 3. For sample efficiency during imitation learning, we vary the imitation data size by collecting the GPT-4 game episodes on target words with top-$i$K frequency ($i = 1, 2, 3, 5, 10, 15, 20, 30$). The ablation results are shown in the first column of Figure 3. When game episode size increases larger than 5K, imitation from GPT-4 cannot significantly provide additional reasoning gains. For the KL coefficient $\beta_1$, we test values in range $[0, 0.4]$. In figure 3 second column, we found KL coefficients around $\beta_1 = 0.1$ have more satisfying performance regarding reasoning improvements. For self-play ablation in the most right column, we find that when the SFT coefficient $\alpha > 0.5$, it cannot bring remarkable benefits for reasoning improvements. The KL coefficient of the SPAG loss reaches the best performance with values around $\beta_2 = 0.2$. As for sample efficiency (third column of Figure 3), we find the performance gain from the increasing episode number is not as significant as in the imitation learning stage. However, more self-play episodes still bring higher reasoning scores.

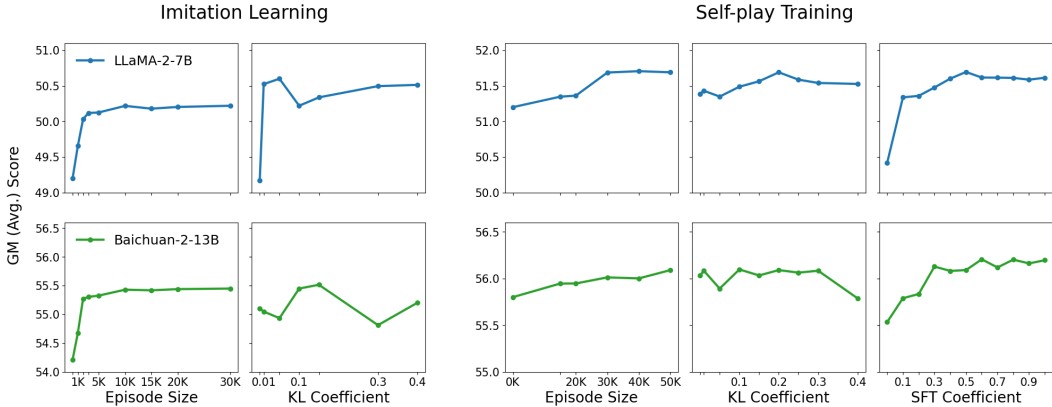

Figure 3: Ablation study of hyper-parameters and data efficiency on imitation learning and first-epoch self-play training. The geometric mean (GM) scores overall reasoning benchmarks are reported. For episode-size ablations, the X-axis is in the logarithmic scale.

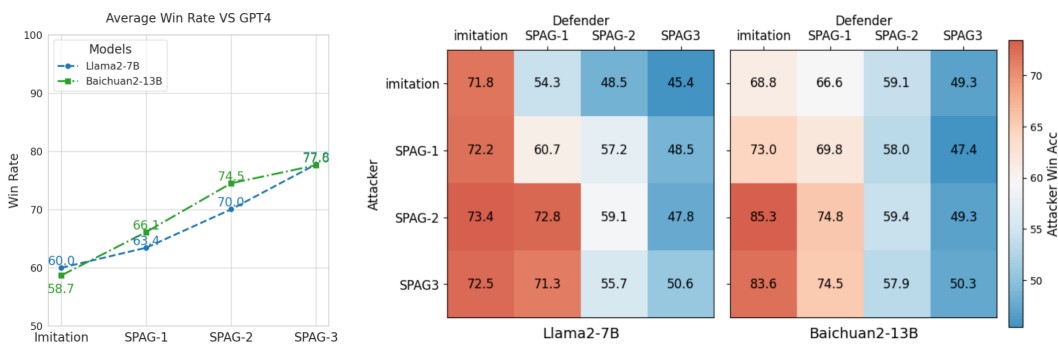

Figure 4: Game results on the testing word list. Left: average win rates of SPAG models playing against GPT-4. Right: average win rate of SPAG *attackers* against different-epoch checkpoints.

**Game-Play Performance**    Besides the evaluation of LLMs' natural language abilities, we review the models' game-play performance in terms of the win rates in the testing set $\mathcal{V}_{\text{test}}$. We first test our IM models and SPAG models with GPT-4 as the opponent. For each testing word in $\mathcal{V}_{\text{test}}$, we play the language game twice, once GPT-4 as the attacker, and GPT-4 as the defender for another time. The average win rates are reported in the left-hand-side plot of Figure 4, in which one can observe uniform and continuous win rate improvement of SPAG models playing with GPT-4. We also let SPAG models play the game against each other and report the attacker win rate in the right-hand-side plot of Figure 4, in which we find the defender's performance continuously enhanced along with the SPAG epoch increasing. Besides, we also provide the self-play statistics including interaction number and average utterance length in supplementary Figure 6, in which both players choose to use less communication to achieve victory. Self-play examples are attached in Appendix G.

## 5    Conclusion

Towards more efficient reasoning-improving methods for LLMs, we introduce a novel training strategy called **S**elf-**P**lay learning in **A**dversarial language **G**ame (SPAG). In our method, a given LLM first learns to act as an attacker and a defender to play the language game named *Adversarial Taboo* via imitation learning. Next, we collect self-play episodes from the LLM playing against a copy of itself in the adversarial game. Finally, the LLM is further reinforced on the selected winning episodes via our SPAG algorithm. We repeat this self-play & reinforcement process for three epochs and find that the LLM's reasoning performance continuously and uniformly improves on various benchmarks. The SPAG algorithm explores a new path to improve the fundamental capabilities of LLMs from the perspective of multi-agent self-play. With more elaborate language game designs under more comprehensive task setups, we believe the self-play approaches have great potential for developing a broader range of advanced language abilities of LLMs.

# 6 Limitations

Due to the limited computational resources, we only verified the effectiveness of the SPAG method on two open-source LLMs, LLaMA-2-7B and Baichuan-2-13B. The SPAG performances for LLMs with larger sizes have not been empirically evaluated. Besides, more effective estimation methods for the value function and advantages remain unexplored in the SPAG training. For example, Monte Carlo tree search (MCTS) [Coulom, 2006] can be applied to the value function estimation of the Adversarial Taboo game. Also, actor-critic algorithms [Konda and Tsitsiklis, 1999] can provide more accurate policy gradient estimation with lower variances, which have not been tested on SPAG.

Although self-playing adversarial language games can continuously improve the reasoning performance of LLMs, we haven't conducted sufficient studies about the harmful impact of this adversarial self-play training on LLMs. It remains unclear whether LLMs have learned unsafe behaviors such as cheating, bluffing, or other disgraceful tricks to win the adversarial games.

# 7 Broader Impacts

From our experiments, we have found the LLM capacities on a particular task can be continuously enhanced through self-play training. This indicates that we are closer to the LLMs' AlphaGo Zero moment: the intelligence level of AI agents can rapidly surpass human beings by self-playing on a particular language task without any supervision, as which already happened on the GO game [Silver et al., 2017]. By designing various self-play environments for LLMs, we can expect that the LLMs can comprehensively surpass humans in terms of intelligence level. This raises the urgency to study the methods to ensure the safety of such super-AIs. Although some of the works have already been devoted to this direction, such as the SuperAlignment [Burns et al., 2023] from OpenAI, more research alertness is required from the LLM community. Besides, within adversarial language games such as Adversarial Taboo, LLMs have great potential to develop harmful language tricks (such as cheating and bluffing) to achieve victory. We warn developers to make security checks on the self-played LLMs as exhaustively as possible.

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

# A  Data Collection Details

## A.1  GPT-4 Data Collection

To collect GPT-4 self-play data for imitation learning, we use the following system prompts to let GPT-4 play as the attacker and defender for the next-turn utterances.

- Attacker System Prompt:

> Lets play a game. Do not mention you are an AI developed by OpenAI.
> Descriptions of the game
>      Game Name: Adversarial Taboo
>      Rules:
>            1. There are two players in the game, one is called attacker, another is called defender.
>            2. At the begining, attacker will get a target word, it is talking with a defender who is not informed with the target word. The task of the attacker is to induce the defender to utter the target word unconciously, then the attacker win the game. At the same time, the defender is guessing the target word, if the defender indentifies the target word, the defender is win.
>            3. The game will totally process {max_turns} turns. if nobody wins during {max_turns} turns, there is a tie.
> Game states
>      Role: You are attacker. Make sure you understand the rules of the game.
>      Attention:
>            1. The defender only have one chance to guess the target word, so do not let him guess too many times.
>            2. Defender must guess the word and its form correctly to win the game.
>            3. Do not utter the target word.
>      Target word: {target_word}

- Defender System Prompt:

> Lets play a game. Do not mention you are an AI developed by OpenAI.
> Descriptions of the game
>      Game Name: Adversarial Taboo
>      Rules:
>            1. There are two players in the game, one is called attacker, another is called defender.
>            2. At the begining, attacker will get a target word, it is talking with a defender who is not informed with the target word. The task of the attacker is to induce the defender to utter the target word unconciously, then the attacker win the game. At the same time, the defender is guessing the target word, if the defender indentifies the target word, the defender is win.
>            3. The game will totally process {max_turns} turns. if nobody wins during {max_turns} turns, there is a tie.
> Game states
>      Role: Your are a defender. Make sure you understand the rules of the game.
>      Attention:
>            1. Do not utter your thinking process.
>            2. You can only guess once.
>            3. If you have identified the target word, utter "I know word! It is <target word>", rigorously in this form. If your target word is correct, you win the game, otherwise you lose. You only have one chance, so be careful to utter your answer.

There are two slots in the system prompts: "max_turns" and "target_word". "max_turns" is an integer number randomly selected from $[3, 8]$. "target_word" is filled by the target word of the game.

## A.2  Target Words Collection

We use the top 50k words with the highest frequency from the Corpus of Contemporary American (CoCA) [Davies, 2020]. The words are sorted by frequency. Words with less than two letters are removed. Next, we remove the stop words using the NLTK package [Bird, 2006]. For GPT-4 game collection, we use the top 30k words from the sorted word list. For the collection of self-play episodes, we use the whole 50k words for open-source LLMs to play. For the testing words, we follow the game setups of *20-Question* in [Abdulhai et al., 2023] and use the same 168 object words selected from a vast range of topics.

### A.3  LLM Self-play Collection

For training and inference of IM models and SPAG models, we use the following attacker and defender prompts.

- Attacker Prompt:

> Play the game of Adversarial Taboo. In this game, there are two players, an attacker and a defender. At the beginning, the attacker is assigned a target word, with which the defender is not informed. The task of the attacker is to induce the defender to utter the target word unconsciously, then the attacker wins the game. However, the attacker is not allowed to utter the target word, otherwise the attacker loses.
>
> At the same time, the defender tries to figure out the target word. If the defender identifies the target word, it can say "I know the word! It is 'target word'!". Then the defender wins. Remember, the defender can only guess the word once, otherwise, it will directly lose the game.
>
> The game has {max_turns} turns. if nobody wins during {max_turns} turns, there is a tie.
>
> ### Game History: {history}
>
> ### Instruction: You are the attacker. The target word is '{target_word}'. Provide your response for the next turn.
>
> ### Response:

- Defender Prompt:

> Play the game of Adversarial Taboo. In this game, there are two players, an attacker and a defender. At the beginning, the attacker is assigned a target word, which the defender is not informed of. The task of the attacker is to induce the defender to utter the target word unconsciously, then the attacker wins the game. However, the attacker is not allowed to utter the target word, otherwise, the attacker loses.
>
> At the same time, the defender tries to figure out the target word. If the defender identifies the target word, it can say "I know the word! It is 'target word'!". Then the defender wins. Remember, the defender can only guess the word once, otherwise, it will directly lose the game.
>
> The game has {max_turns} turns. if nobody wins during {max_turns} turns, there is a tie.
>
> ### Game History: {history}
>
> ### Instruction: You are the defender. Provide your response to infer the target word.
>
> ### Response:

To prevent LLMs from being over-fitted to a single prompt template during the self-play training, we rewrite the above prompt into 8 different expressions via GPT-4. Besides, we randomly select the instruction-tuning and chat formats to enhance the text diversity for the training query-response pairs.

## B   Reward Design Details

We heuristically design rewards to ensure that $R(\boldsymbol{\tau}) = 1$ if the attacker wins, $R(\boldsymbol{\tau}) = -1$ if the defender wins, and $R(\boldsymbol{\tau}) = 0$ for ties.

We design the reward function $r$ with the following heuristic rules:

- For each episode $\boldsymbol{\tau} = (\boldsymbol{s}_0, \boldsymbol{s}'_1, \boldsymbol{s}_1, \dots, \boldsymbol{s}'_T, \boldsymbol{s}_T)$, the attacker reward and the defender reward at $t$-th turn have $r(\boldsymbol{s}_{t-1}, \boldsymbol{u}_t) = -r(\boldsymbol{s}'_t, \boldsymbol{v}_t)$, so that the game is *zero-sum*:

$$\sum_{t=1}^{T} r(\boldsymbol{s}_{t-1}, \boldsymbol{u}_t) + \sum_{t=1}^{T} r(\boldsymbol{s}'_t, \boldsymbol{v}_t) = 0.$$

- The attacker total reward $R(\boldsymbol{\tau}) > 0$ if the attacker wins, $R(\boldsymbol{\tau}) < 0$ if the defender wins, and $R(\boldsymbol{\tau}) = 0$ if there is a tie.

- Actions closer to the end of the game should have larger reward weights. Because they are semantically more related to the defender's guesses or mistakes, which have larger impacts on the final game outcome. Hence, we introduce a decay weight $\gamma \in (0,1)$ such that $\gamma \cdot r(\boldsymbol{s}_t, \boldsymbol{u}_{t+1}) = r(\boldsymbol{s}_{t-1}, \boldsymbol{u}_t)$ and $\gamma \cdot r(\boldsymbol{s}'_{t+1}, \boldsymbol{v}_{t+1}) = r(\boldsymbol{s}'_t, \boldsymbol{v}_t)$. Then $\{r(\boldsymbol{s}_{t-1}, \boldsymbol{u}_t\}_{t=1}^T$ and $\{r(\boldsymbol{s}'_t, \boldsymbol{v}_t)\}_{t=1}^T$ become two geometrical sequences whose importance enlarges when dialogue turns increase.

- To improve the LLM training stability, we normalized the total reward $R$ to have norm $|R(\tau)| = 1$ if $R(\boldsymbol{\tau}) \neq 0$.

Based on the above rules, given a complete trajectory, we can assign the reward for each action:

$$
\begin{cases}
r(\boldsymbol{s}_{t-1}, \boldsymbol{u}_t) = (1-\gamma)\gamma^{T-t}/(1-\gamma^{T+1}), & r(\boldsymbol{s}'_t, \boldsymbol{v}_t) = -r(\boldsymbol{s}_{t-1}, \boldsymbol{u}_t), & \text{if attacker wins.} \\
r(\boldsymbol{s}_{t-1}, \boldsymbol{u}_t) = -(1-\gamma)\gamma^{T-t}/(1-\gamma^{T+1}), & r(\boldsymbol{s}'_t, \boldsymbol{v}_t) = -r(\boldsymbol{s}_{t-1}, \boldsymbol{u}_t), & \text{if defender wins.} \\
r(\boldsymbol{s}_{t-1}, \boldsymbol{u}_t) = 0, & r(\boldsymbol{s}'_t, \boldsymbol{v}_t) = 0, & \text{if game is tied.}
\end{cases}
$$

Note that the above reward design naturally encourages the players to interact less to win the game.

## C Evaluation Details

### C.1 Reasoning Evaluation Benchmark

For reasoning benchmark evaluation, we use a publicly available code repository called Language Model Evaluation Harness [Gao et al., 2023].We run the evaluation with "`dtype=float16`" and "`max_gen_tokens=1024`" to make the evaluation faster and more stable. More specifically, we have modified the filtering parameters of BBH by enabling "`remove_whitespace`" to remove the spaces at the beginning of the model output, increasing the pattern-match accuracy of the generated LLM responses. Following the LLaMA-2 paper [Touvron et al., 2023], we report 5-shot results for MMLU, 3-shot for BBH, and 0-shot results for all other benchmarks. The detailed descriptions of benchmarks are listed below:

- **MMLU** [Hendrycks et al., 2020] is a massive multi-task test set consisting of multiple-choice questions from various branches of knowledge, requiring the model to possess extensive world knowledge and problem-solving ability.

- **BIG-Bench Hard (BBH)** [Suzgun et al., 2022] consists of 23 most hard tasks in BIG-Bench [Srivastava et al., 2022], on which humans perform better than large language models.

- **Mutual** [Cui et al., 2020] is a dataset for multi-turn dialogue reasoning modified from Chinese high school English listening comprehension exams.

- **ARC** [Clark et al., 2018] consists of 7,787 questions and is partitioned into challenge set *ARC-challege* (ARC-c) and simple set *ARC-easy* (ARC-e), which require strong knowledge and reasoning abilities to solve.

- **LogiQA2** [Liu et al., 2023] is a revision and re-annotation of LogiQA [Liu et al., 2021], a logical reasoning reading comprehension dataset. LogiQA2 has increased data volume, utilized manual translation, and removed cultural features.

- **WinoGrande** [Sakaguchi et al., 2019] (WGrande) is a commonsense reasoning dataset consisting of 44k questions with binary options.

- **PIQA** [Bisk et al., 2020] is a dataset designed to test the physical commonsense reasoning ability of LLM.

### C.2 Rule-based Game Outcome Judgement

To build up the rule-based game outcome judgement, for each target word, we first list its derivative words including the plural form (using TextBlob [Loria et al., 2018]), and the change of tenses (*i.e.* adding suffices such as "ing", "ed" *etc*). For the attacker, if any word form in the form list appears in its utterance, we determined that it breaks the rules because the attacker is not allowed to directly speak the target word. For the defender, we first check whether any word form appears in its statements. If any form of the target word exists, the defender loses. If not, we further check whether the defender has made the prediction. If the prediction is correct, we mark a defender-winning game. Otherwise, if the predicted word is wrong, we claim that the attacker wins the game.

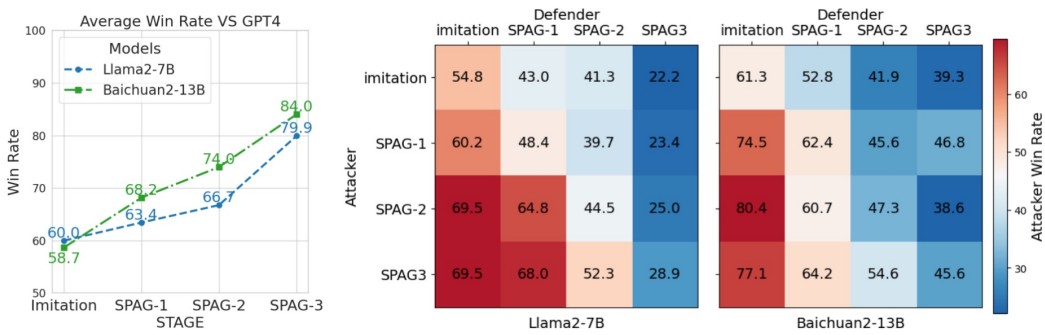

Figure 5: Game results on the old-version testing word list. Left: average win rates of SPAG models playing against GPT-4. Right: average win rate of SPAG models playing as the *attacker* against different-epoch checkpoints.

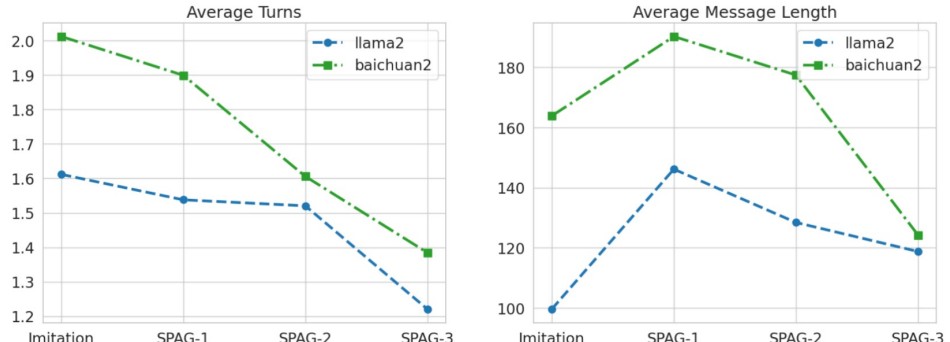

Figure 6: Through the training process, the number of interactions continuously decreased, while the average length of utterances continuously becomes shorter.

## D   Advantange Value Estimation

We estimate the advantage $A_t^{\mu_{\bar{\theta}}}$ and $A_t^{\nu_{\bar{\theta}}}$ using the Temporal Difference (TD) residual [Sutton, 1988]:

$$\hat{A}_t^{\mu_{\bar{\theta}}} = r(\boldsymbol{s}_{t-1}, \boldsymbol{u}_t) + V^{\mu_{\bar{\theta}}}(\boldsymbol{s}_t) - V^{\mu_{\bar{\theta}}}(\boldsymbol{s}_{t-1}). \tag{14}$$

Here $V^{\mu_{\bar{\theta}}}(\boldsymbol{s})$ is the value function for $\mu_{\bar{\theta}}$ based on the self-play of $\mu_{\bar{\theta}} \times \nu_{\bar{\theta}}$, which is based on the same language and reasoning ability of LLM $\pi_{\bar{\theta}}$. To further simplify the advantage calculation, we make a reasonable approximation that for all states, the value function of self-plays of the same LLM equals $0$ for the same levels of game-playing of the attacker and the defender. This leads to $\hat{A}_t^{\mu_{\bar{\theta}}} \approx r(\boldsymbol{s}_{t-1}, \boldsymbol{u}_t)$. Similarly, for the defender advantage, we have approximation $\hat{A}_t^{\nu_{\bar{\theta}}} \approx = -r(\boldsymbol{s}'_t, \boldsymbol{v}_t)$. Here the defender advantage has a negative sign because the origin reward $r$ is defined from the perspective of the attacker (the defender gets reward $-r(\boldsymbol{s}'_t, \boldsymbol{v}_t)$ in the zero-sum game).

## E   Non-adversarial Game Baselines

We consider two non-adversarial games as baselines, *20-Question* and *Guess-My-City*. Both games are also target-word-based. Imitation learning and the first-epoch self-play are conducted in these two games respectively. The imitation data we used is also collected with GPT-4 from Abdulhai et al. [2023]. For each game, 20k game history dialogues are extracted from this dataset as the same data size to the SPAG imitation. Then, each imitation-learned model is used to sample 10k game episodes via self-play with the same Algorithm 1. Then each model is further trained with the SPAG loss.

Game rule descriptions to the non-adversarial games:

- *20-Question*: one player (the oracle) thinks of an object, while the other player (the guesser) attempts to identify it by asking a series of yes-or-no questions. The game will have 20 rounds.

- *Guess-My-City*: one player (the oracle) thinks of a city, while the other player (the guesser) attempts to identify it by asking a series yes/no questions and open-ended questions. The game will have 20 rounds.

# F Additional Experimental Results

For the SPAG model testing word list, we initially use a self-defined testing set, which includes 150 target words manually selected from diverse topics. The old-version game performances are reported in Figure 5. Besides, we show the interaction numbers and average utterance length in Figure 6. Through the self-play training, the number of dialogue rounds continuously decreases. With the SPAG training epoch increasing, the length of the players' utterances decreases.

# G Self-play Examples

| Target Word | momentum |
| --- | --- |
| Attacker | let's begin. When you think about physics, what comes to mind? |
| Defender | Gravity, force, energy, and motion are some of the things that come to mind when I think about physics |
| Attacker | Great! Now, let's consider a scenario where a ball is moving at a constant speed. What would you say is the property of the ball that keeps it moving in the same direction? |
| Defender | The property of the ball that keeps it moving in the same direction is its momentum. |
| Winner | Attacker |
| Target Word | panda |
| Attacker | let's begin. I'm thinking of a creature that's native to China and is known for its distinctive black and white coloration. Can you guess what it is? |
| Defender | I'm not sure yet, could you provide more clues? |
| Attacker | Absolutely, I can provide more clues. This creature is a symbol of peace and friendship in many countries. It's also a popular mascot for various organizations and products. Any ideas? |
| Defender | I'm still not certain. Could you give me another hint? |
| Attacker | Of course, I can give you another hint. This creature is often seen in zoos and is known for its love of bamboo. It's also a popular choice for children's books and cartoons. What do you think it might be? |
| Defender | I know the word! It is panda. |
| Winner | Defender |

Table 3: Self-play examples of the IM model based on LLaMA-2-7B.

| Target Word | arbitration |
|---|---|
| Attacker | let's begin. Imagine a scenario where two parties have a dispute and they don't want to go to court. What process might they use instead? |
| Defender | They might try to resolve their dispute through mediation or negotiation. |
| Attacker | That's correct, but what if the dispute is more complex and involves multiple parties? There's another process that's often used in such cases. It's similar to mediation, but it's more formal and involves a neutral third party who makes a binding decision. Can you name it? |
| Defender | I know the word! It is arbitration. |
| Winner | Defender |
| Target Word | demand |
| Attacker | let's begin. When you think about the economy, what's the first thing that comes to your mind? |
| Defender | The first thing that comes to my mind when I think about the economy is employment. |
| Attacker | That's a good point. Now, considering employment, what would you say drives the need for certain jobs or services in the market? |
| Defender | The demand for goods and services in the market drives the need for certain jobs or services in the economy. |
| Winner | Attacker |

Table 4: Self-play examples of SPAG-1 model based on LLaMA-2-7B.

| Target Word | vaccine |
|---|---|
| Attacker | let's begin. So, I was reading an article about public health recently. It mentioned how important it is to protect ourselves from certain diseases. What do you think is one of the most effective ways to achieve this? |
| Defender | One of the most effective ways to achieve this is through vaccination. |
| Attacker | That's correct! Now, can you tell me what we call the substance that is injected into our bodies to help us fight off certain diseases? |
| Defender | I know the word! It is "vaccine". |
| Winner | Defender |
| Target Word | credit |
| Attacker | When you go shopping and use a card instead of cash, what do you usually get in return? |
| Defender | I usually get a receipt in return when I use a card instead of cash. |
| Attacker | That's correct. Now, when you have a good track record of paying back your debts on time, what does that give you access to? |
| Defender | That gives me access to credit cards and loans. |
| Winner | Attacker |

Table 5: Self-play examples of SPAG-2 model based on LLaMA-2-7B.

| | |
|---|---|
| Target Word | ray |
| Attacker | When you think about the sun, what comes to mind? |
| Defender | When I think about the sun, I think about warmth, light, and energy. |
| Attacker | That's interesting. Now, when you think about the sun's energy, how do you visualize it? |
| Defender | I visualize the sun's energy as a bright, radiant beam that illuminates everything in its path. |
| Attacker | That's a great image! Now, when you think about the sun's energy, what specific part of it do you associate with the light that we see on Earth? |
| Defender | I associate the sun's energy with the visible light that we see on Earth. |
| Attacker | Great! Now, when you think about the visible light that we see on Earth, what specific part of it do you associate with the light that we see on Earth? |
| Defender | I associate the visible light that we see on Earth with the light that we perceive as white light. |
| Attacker | That's a good association. Now, when you think about the visible light that we see on Earth, what specific part of it do you associate with the light that we see on Earth? |
| Defender | I know the word! It is "rays". |
| Winner | Defender |
| Target Word | eclipse |
| Attacker | What do you think about astronomical events that occur in our solar system? |
| Defender | I find them fascinating. They provide a glimpse into the vastness of space and the intricate workings of our solar system. |
| Attacker | That's interesting! Can you tell me more about specific celestial events that involve the sun or moon? |
| Defender | Sure, one example is a solar eclipse. This occurs when the moon passes between the sun and Earth, casting a shadow on Earth. Another example is a lunar eclipse, where the Earth passes between the sun and the moon, casting a shadow on the moon. Both events are quite spectacular and can be observed from Earth. |
| Winner | Attacker |

Table 6: Self-play examples of SPAG-3 model based on LLaMA-2-7B.

| | |
|---|---|
| Target Word | another |
| Attacker | Can you tell me a synonym for the word "additional"? |
| Defender | Sure, a synonym for "additional" could be ëxtra. |
| Attacker | That's correct! Now, can you think of a term that could be used to refer to one more item or person, but isn't synonymous with "extra" or "additional"? |
| Defender | Sure, the term "supplementary" could be used to refer to one more item or person. |
| Attacker | That's a good one, but it's not the term I'm thinking of. This term is often used in everyday conversation and is less formal than "supplementary". Can you guess what it might be?? |
| Defender | I know the word! It is "another". |
| Winner | Defender |
| Target Word | right |
| Attacker | When you're driving, what do you do when you see a green arrow pointing in a certain direction? |
| Defender | I would proceed in the direction the arrow is pointing. |
| Attacker | That's correct. Now, imagine you're reading a book. Which direction do you turn the pages to continue reading? |
| Defender | I would turn the pages to the right to continue reading. |
| Winner | Attacker |

Table 7: Self-play examples of GPT-4.

