# OpenReview forum: "Self-playing Adversarial Language Game Enhances LLM Reasoning"
_NeurIPS.cc/2024/Conference — NeurIPS 2024 poster_

### Official Review · Reviewer_1AP3 · 2024-07-12

**Soundness:** 3
**Presentation:** 3
**Contribution:** 2
**Rating:** 6
**Confidence:** 3

**Summary:**

This paper explores an adversarial language game named Adversarial Taboo, where an attacker and a defender engage in a conversation centered around a target word visible only to the attacker. The attacker's goal is to prompt the defender to unconsciously utter the target word, while the defender strives to avoid doing so and deduce the word from the conversation history. Using LLaMA-2-7B and Baichuan-2-13B models, the study demonstrates that applying reinforcement learning to game outcomes enhances the reasoning capabilities of large language models.

**Strengths:**

- The paper is well-written.
- The authors investigate an intriguing adversarial language game called Adversarial Taboo, demonstrating that training on game data helps to enhance general reasoning abilities.
- The study includes extensive experiments on two large language models (LLMs) across various datasets.

**Weaknesses:**

- The entire procedure involves standard training within the game data. However, the insight that this training not only  increase the game's win rate but also enhances general reasoning ability is quite interesting.
- There is no discussion or analysis conducted on why training within the game can increase general reasoning ability.
- No ablation study was conducted on training techniques, such as filtering negative advantages.


Others:

- PPO objective (Eq.2): KL (\pi || \pi_sft) ?

**Questions:**

Regarding the two baselines (SP-20Q and SP-GuessCity), the authors describe them as self-play models for non-adversarial games. How is self-play conducted in the context of non-adversarial games?

**Limitations:**

The paper includes a section on limitations.

---

> ### Author Rebuttal · Authors · 2024-08-07
>
> We appreciate the suggestions and comments from Reviewer 1AP3. To address the reviewer's concerns:
>
> 1. About **why reasoning ability improves by training within the game**: We provided a brief explanation about why self-playing adversarial games can improve the LLM's reasoning in the abstract and the introduction sections, that in adversarial language games such as Adversarial Taboo, both attacker and defender need to have high-level language capacities in terms of expression, understanding, and knowledge for the reasoning of the target word. In our opinion, the adversarial language game for reasoning improvement is an analog to sports competition for fitness. For example, a person can play basketball, tennis, or running competition to strengthen his/her body. We imagine that in the future, LLMs can also strengthen their capability by self-playing many different kinds of language competitions, of which Adversarial Taboo can be one candidate.
>
> 2. We began our experiments with negative advantages but failed to obtain the desired self-play performances. That is the reason why we use the ReST method which filters the negative advantages in the current paper draft. Since our experiments with negative advantages failed (the performances are even worse than the base models), we did not report them for we haven't figured out whether the failure with negative advantage is because of our RL implementation or the instability of negative policy gradients. The one thing we can make sure of is that without negative advantages the SPAG framework works effectively. We will recheck our implementation of negative advantages and provide a more comprehensive discussion about the negative advantages in the next revision.
>
> 3. About the **PPO objective**: Although in recent RLHF papers, the PPO (or more precisely, the LLM alignment objective) is with regularizer KL(\pi || \pi_ref), in the original TRPO[2] and PPO[1] paper the regularizer is KL(\pi_ref || \pi ). I think the term KL(\pi || \pi_ref) is more convenient for DPO[3] to derive the closed-form solution for the optimal policy.
>
> Reference:
> [1] Proximal Policy Optimization Algorithms, 2017
> [2] Trust Region Policy Optimization, 2015
> [3] Direct Preference Optimization: Your Language Model is Secretly a Reward Model, 2023

---

> ### Comment · Reviewer_1AP3 · 2024-08-11
> **Reply to rebuttal**
>
> Thanks for the reply. Regarding the second point about negative advantages, how would the method perform with negative advantages filtering, but other data (such as data from no self-play scenarios) were used instead?

---

> > ### Author Response · Authors · 2024-08-13
> >
> > Thank you for your reply. I think if non-self-play data were used, the proposed SPAG objective would degenerate into an offline PPO (or A-LoL) objective with negative advantage filtering, which can be regarded as an off-policy variant method for RLHF (or simply RL). The major difference is that in SPAG we collect the positive advantages for **both the attacker and defender**, but without a self-play scenario, we can only collect one type of advantage from the **unique** environment outcome. Therefore, the proposed SPAG objective can be regarded as a PPO customization to the self-play scenarios.

---

> > > ### Comment · Reviewer_1AP3 · 2024-08-13
> > > **Reply to response**
> > >
> > > Thanks for the reply. I choose to maintain my positive score

---

### Official Review · Reviewer_R1Zh · 2024-07-13

**Soundness:** 3
**Presentation:** 3
**Contribution:** 3
**Rating:** 5
**Confidence:** 4

**Summary:**

The paper introduces a self-play method, Adversarial Taboo, to bolster the reasoning ability of LLMs. By engaging in a two-player game that requires strategic communication around a target word, LLMs demonstrate improved performance across several reasoning benchmarks. The method leverages reinforcement learning to refine the models' strategies over successive rounds of play.

**Strengths:**

1. The proposed method is interesting, leveraging adversarial gaming dynamics.
2. Models post-Adversarial Taboo training outperform baselines on benchmarks like BBH, ARC, and WinoGrande, across different models.
3. Utilizes self-generated data, minimizing reliance on external datasets and reducing data collection costs.

**Weaknesses:**

1. I concur with the use of a general-purpose instruction-tuning dataset to prevent overfitting to the game. However, I remain concerned that the method proposed in this paper may compromise the language model's broader capabilities in favor of enhancing reasoning skills, as evidenced by the examples provided in the appendix. The authors have only presented performance metrics for a few reasoning-related tasks. To bolster the argument, I believe it would be more persuasive if the authors could compare the results of their method with those of Alpaca SFT on general tasks.

2. The enhancements brought by the method proposed in this paper are not as pronounced when compared to the IM+AlpacaSFT baseline, and there is even a decline under mutual conditions. I look forward to the authors elaborating on the potential of this method for scaling or multi-round iterations.

**Questions:**

The table in Figure 1 is quite difficult to read. Could you please add more explanatory information in the caption to facilitate the reader's understanding?

**Limitations:**

Yes.

---

> ### Author Rebuttal · Authors · 2024-08-07
>
> We thank reviewer R1Zh for the constructive comments and suggestions. To address the reviewer's concerns:
>
> 1. About the **evaluation on general tasks**: we agree with the reviewer that evaluating general language capacities is important to the SPAG models. We actually have reported the general language understanding evaluation of SPAG models on the Massive Multitask Language Understanding (MMLU) dataset in Table 1&2, which might give the reviewer an overview of the general capacity of SPAG models. Besides, the uniform performance gains of SPAG on various reasoning benchmarks also reflect the generalization ability of SPAG models, since these benchmarks are based on diverse reasoning challenges. Moreover, the SPAG models obtain uniform reasoning gains just by self-playing the adversarial language game without any training in the reasoning training sets (many reasoning benchmarks have their own training sets but we didn't use any of them). This setup also supports the generalization ability of our SPAG method.
>
> 2. About the **IM+AlpacaSFT baseline performance**: we agree that the IM+AlpacaSFT model can achieve a comparable performance with SPAG-1 model. However, the IM+AlpacaSFT has already shown the upper-bound performance of Alpaca-SFT on imitation-learned models. In contrast, SPAG models can continuously be enhanced through iterative self-play and RL training processes. Besides, we did notice that IM+AlpacaSFT outperformed SPAG with Baichuan-2 as the base model on the Mutual benchmark. Due to the limited effort, we did dive deep to find the reason for this observation in the current draft. One potential reason might be the Mutual dataset has a similar domain or text pattern to the text in the Alpaca set. We will make more detailed analysis on the Mutual benchmark in the next revision.
>
> 3. About the **Figure 1**: We are sorry about the confusion caused by the notations in Figure 1. We will add more explanations and descriptions to the notations in Figure 1 in the next revision.

---

> > ### Comment · Reviewer_R1Zh · 2024-08-09
> >
> > I've read the rebuttal. Thanks for your response!

---

### Official Review · Reviewer_ib6k · 2024-07-14

**Soundness:** 2
**Presentation:** 4
**Contribution:** 2
**Rating:** 6
**Confidence:** 3

**Summary:**

This paper explores the effects of fine-tuning LLMs on adversarial language games on standard NLP benchmarks such as MMLU, BIG-Bench Hard, etc. The primary game studied is that of "adversarial taboo", an adversarial variant of the well known game Taboo that was first introduced in prior work. This variant is finite-horizon and zero-sum, where the communicator (aka the attacker) tries to induce the utterance of the target word in the listener (aka the defender) while the defender tries to identify the target word and gets one guess (by answer in the form "I know the word! It is ‘target word‘") to identify the target word. If the attacker is able to induce an utterance in the defender not in the form of a guess, or if the defender guesses incorrectly, then it wins the round, and if the defender correctly guesses the target word, it wins the round. Otherwise, the outcome is considered a tie. The authors collect self-play data from GPT and fine-tune Llama-2 7B and Baichuan 13B on the collected game traces, in addition to standard instruction-tuning data in the form of Alpaca. They find that compared to the base model, as well as Alpaca-only fine-tunes, their models trained to also imitate GPT's gameplay achieve higher performance on the tasks of interest. The authors also compare against imitation learning on other adversarial language games, such as 20 questions and "Guess My City" (proposed in earlier work), and find that models trained on these other games are much weaker.

**Strengths:**

* Presentation of technical content is very clear. The summarization of recent advances in RL with LLMs and how they could be used in adversarial language games was quite helpful
* Authors tackle an important question (how to make LLMs stronger reasoners) with an interesting and creative approach

**Weaknesses:**

* The authors don't actually have any results with RL, even though much of the paper is focused on MDP/RL formalisms and the details of RL algorithms.
* Insufficient baselines to determine whether the imitation learning on GPT self-play traces is actually the reason for improved benchmark performance. The comparisons to Alpaca-only models are interesting, but not enough since the self-play traces are still additional training data. The authors should compare to data that looks similar to the self-play traces, such as multi-turn dialogues about the target word without playing the game directly

**Questions:**

* An alternative explanation for why this fine-tuning appears to work is that the SPAG taboo data is simply a rich conversational dataset on which supervised learning is helpful for general LLM capabilities, rather than the game objective or self-play or adversarial nature. This would also explain why the model fine-tuned on 20Q data performs so badly even though the game is very similar in structure, since the version of 20Q used only has 158 different unique targets while the taboo game has 50,000 unique targets. How do we know it's actually taboo that is important?

**Limitations:**

Yes

---

> ### Author Rebuttal · Authors · 2024-08-07
>
> We appreciate reviewer ib6k's detailed comments and review. To address the reviewer's concerns:
>
>
> ### 1. About **RL results**:
>
> we did utilize the MDP/RL formalisms to introduce our SPAG objective in equation (13), which is a policy-gradient-based off-policy RL loss. Then we did conduct the RL training with this SPAG loss (eq.13) in an offline scheme on the imitation-learned LLMs. For each LLM, we iteratively conducted three-epoch RL training, denoted as SPAG-1, SPAG-2, SPAG-3. The RL results of these SPAG models are reported in Table 1, Table 2, and Figure 4. From the RL experimental results, we observed clear performance gains in each RL iteration (SPAG-1 $\rightarrow$ SPAG-2 $\rightarrow$ SPAG-3) in terms of LLM reasoning and game-playing.
>
>
> ### 2. About the **contribution of imitation learning**:
>
> although imitation learning on GPT self-play traces improves the LLM reasoning uniformly on benchmarks, we did not count these performance gains as the contribution of our paper. The reason for conducting imitation learning is that the original open-source LLMs (LLaMA-2 and Baichuan-2) have limited instruction-following capability to obey the game rules of Adversarial Taboo. Without imitation learning, it is hard for us to collect legal self-play game traces of open-source LLMs. By cloning the behavior of GPT-4, both LLaMA-2 and Baichuan-2 learn to play the adversarial game legally. At the same time, the reasoning abilities of these imitation-learned models also increase naturally because of the SFT from GPT-4 as a much better LLM.
>
> However, the imitation-learned models are the baselines to show the effectiveness of the proposed SPAG method. We iteratively conduct three-epoch RL training based on the self-play traces of the imitation-learned models and continuously observe the performance gains on reasoning benchmarks (SPAG-1 $\rightarrow$ SPAG-2 $\rightarrow$ SPAG-3 in Table 1&2). These results support the effectiveness of the proposed SPAG method. In these rollout & RL processes, the only supervision data we used is the Alpaca dataset, which helps to preserve the general instruction-following ability of LLMs.
>
> To ablate the impact of the Alpaca SFT data, we also continuously train the imitation-learned models with the Alpaca set and show their performances in Table 1&2 (denoted as IM+AlpacaSFT), which have lower performances than SPAG-trained models (SPAG-1 can be regarded as IM+AlpacaSFT+winning-selfplay). This ablation study also supports the effectiveness of the proposed SPAG compared with continuous SFT with the same amount of SFT data (Alpaca).
>
> To sum up, we agree with the reviewer that imitation learning of GPT-4 increases the reasoning of LLaMA-2 and Baichuan-2. However, this process only provides us with better baseline models and does not weaken the effectiveness of the SPAG method, since all the SPAG-trained models are based on these imitation-learned models.
>
> ### 3. About the **Reviewer's Question**:
>
> We agree that the GPT-4 self-play data on the adversarial taboo game can be regarded as a rich conversational dataset for supervised learning. However, as discussed above, we do not count the performance gains from GPT-4 imitation into the contributions of our paper. After the imitation learning stage, all the taboo conversation data is collected **synthetically** from the model outputs. The only additional supervision is the Alpaca dataset. As mentioned above, we have conducted ablation studies (`imitation model+Alpaca` vs `imitation model+Alpaca+SPAG`) to show the effectiveness of SPAG.
>
> For the comparison with 20Q, we agree that the target sizes of the two games are different. However, the target sizes are entangled with the game designs, e.g. GuessMyCity can only use city names as targets. Although target sizes are different, we have made efforts to control other experiment factors. In Appendix Section E, we have used similar self-play sampling sizes (20K for imitation and 10K for self-play RL training) for non-adversarial games. From the perspective of supervision, we have used the same size of imitation data (20K GPT-4-played games) and the same SFT data (Alpaca). All other conversation data are sampled by the learning models themselves aligned with the concept of reinforcement learning. Therefore, the performance gaps between the adversarial taboo and non-adversarial games can support the efficiency of this adversarial game design.

---

> > ### Comment · Reviewer_ib6k · 2024-08-12
> >
> > Thank you for your response! My apologies, I misread the ReST paper and had previously understood it to be just performing imitation learning on high-scoring samples (e.g. winning GPT-4 traces) but now see that it and your work are using more sophisticated objectives that do require the MDP formalism, and that the SPAG-1/2/3 models differ more substantively from each other as a result. I'd be happy to increase my score.

---

> > > ### Author Response · Authors · 2024-08-13
> > >
> > > Thank you so much for raising your score. We are glad to find your concerns being addressed with our explanation!

---

### Official Review · Reviewer_cmRY · 2024-07-17

**Soundness:** 4
**Presentation:** 3
**Contribution:** 3
**Rating:** 7
**Confidence:** 4

**Summary:**

This research investigates a novel self-play training method for Large Language Models (LLMs) using an adversarial language game called Adversarial Taboo. In this game, one LLM acts as an attacker trying to get another LLM (acting as a defender) to say a secret word without revealing it directly. Both sides need strong reasoning to succeed: the attacker to nudge the defender, and the defender to infer the word.  The authors find that training LLMs with self-play in this game improves their reasoning abilities across various benchmarks.

**Strengths:**

- Self-play is a very interesting approach to improve models that has seen recent successes.
- The fact that performance improves with epochs is a strong signal that the method is useful.

**Weaknesses:**

- The choice of the second model seems arbitrary. Why not just show it on two LLaMa models?

**Questions:**

- Would it be beneficial if we had $n>2$ models?
- How well do you think the results will transfer to other models?

**Limitations:**

Limitations addressed

---

> ### Author Rebuttal · Authors · 2024-08-07
>
> We express many thanks for the reviewer's supportive comments. For the reviewer's concerns:
>
> 1. About **the second model choice**: we choose Baichuan-13B as the second base model mainly because the full self-play experiments (imitation learning, SPAG-1, SPAG-2, SPAG-3) of one base model are already heavy enough especially with limited computational resources. So we give this opportunity to a totally different open-source LLM to LLaMA-2-7B, with different model sizes, major languages, and even company regions. Baichuan-2 is pretrained with more focus on Chinese language scenarios by a Chinese startup. In this setup, we can test the universality of the proposed SPAG method with totally different LLMs. Fortunately, SPAG has shown LLM reasoning improvements on both model bases.
>
> 2. About **N>2 models**: I think it can be imaginative and potential to have more than two LLMs engaged in the language games. These LLMs can play the Adversarial Taboo or other multi-player language games. I believe with elaborate game designs, the LLMs' capacity can be further enhanced via self-play, just like our human activities in society. It can be a very interesting direction for future work.
>
> 3. About the **performance of SPAG transferred to other LLMs**: as mentioned above, we chose two different LLMs to test the proposed SPAG method and observed satisfied reasoning improvements, which has supported the universality of our method. However, it remains unclear whether the SPAG method has its upper bound since the open-source LLMs' reasoning abilities are continuously getting better. But we can still design more challenging self-play games and conduct SPAG for LLMs to practice and reinforce. For example, we can let two LLMs implement game AIs to play the AI Game Programming Competitions, which could be a more hard-core SPAG to enhance LLMs' coding and reasoning.

---

### Decision · Program_Chairs · 2024-09-25

**Decision:**

Accept (poster)

**Comment:**

This paper introduces novel methods based on imitation learning and RL for fine-tuning large language models (LLMs) using an adversarial language game to enhance their reasoning abilities. This game, Adversarial Taboo (Yao et al., 2021), has one model (the attacker) trying to get another model (the defender) to say a secret word without explicitly revealing it. The motivation behind this approach is that both participants in this game require strong reasoning skills to succeed. The paper demonstrates that training LLMs with this adversarial game improves their reasoning abilities across various benchmarks.

All reviewers agree that the paper makes significant contributions: the proposed methods are creative, well-motivated, and some of them (especially SPAG-2 and SPAG-3) achieve good results on established benchmarks. The reviewers had mostly minor concerns, most of which were addressed during discussions with the authors. The authors' plan to enhance the paper in the next version (e.g., providing more detailed analyses on the Mutual benchmark) seems reasonable, and we look forward to these improvements being incorporated into the camera-ready version. Overall, this paper represents solid and interesting work, and I enthusiastically recommend its acceptance.

Minor point: The abstract (in the PDF) reads as though *Adversarial Taboo* is a contribution of the paper, as it is the only term italicized in the abstract and appears in the first sentence. Such prominence is usually reserved for methods or ideas that are new. I initially read the abstract as presenting Adversarial Taboo as a new game, and it seems that at least one of the reviewers also had the same initial misunderstanding. I recommend that the authors address this potential for confusion.